# Exploring Nurses’ Working Experiences during the First Wave of COVID-19 Outbreak

**DOI:** 10.3390/healthcare10081406

**Published:** 2022-07-27

**Authors:** Areti Stavropoulou, Michael Rovithis, Evangelia Sigala, Maria Moudatsou, Georgia Fasoi, Dimitris Papageorgiou, Sofia Koukouli

**Affiliations:** 1Department of Nursing, School of Health and Care Sciences, University of West Attica, 12243 Athens, Greece; esigala@uniwa.gr (E.S.); gfasoi@uniwa.gr (G.F.); dpapa@uniwa.gr (D.P.); 2Department of Nursing, School of Health Sciences, Hellenic Mediterranean University, 71410 Heraklion, Greece; rovithis@hmu.gr; 3Department of Social Work, School of Health Sciences, Hellenic Mediterranean University, 71410 Heraklion, Greece; moudatsoum@hmu.gr (M.M.); koukouli@hmu.gr (S.K.)

**Keywords:** COVID-19, nursing care, nurses’ experiences, qualitative study

## Abstract

During the COVID-19 outbreak, nurses employed in the clinical sector faced a number of difficulties associated with excessive workload, increased stress, and role ambiguity, which impacted nurses themselves and patient care. The aim of the present study was to investigate how Greek hospital nurses working in non-COVID units experienced the virus outbreak during the first wave of the pandemic. A descriptive qualitative research design was applied using a content analysis approach. To recruit the study participants a purposive sampling strategy was used. Ten nurses participated in the study. Data collection was conducted through semi-structured interviews. Content analysis revealed three themes namely, (a) emotional burden, (b) professional commitment, and (c) abrupt changes. Six subthemes were formulated and assimilated under each main theme respectively. Organizational changes, emotional burdens and feelings of fear and uncertainty, appeared to have a crucial effect on nurses and patient care. However, the professional commitment and the nurses’ effort to provide excellent nursing care remained high. Nurses demonstrated that despite the burdens caused by the COVID-19 outbreak, the pandemic era created opportunities for thoroughness and accuracy in nursing care.

## 1. Introduction

In late 2019, the novel COVID-19 virus was identified as a rapidly spreading respiratory disease in Wuhan, China. Since then, the virus has spread worldwide, gaining a pandemic extent, with serious health, economic and social consequences. The latest evidence indicates that by April 2022, approximately 489 million people have been infected and over 6 million deaths have been reported globally [1].

Nurses have been at the forefront of caring for COVID-19 patients since the onset of the pandemic. Various studies highlighted the psychological impact on nurses working in COVID clinics, such as anxiety, stress, depression, and mental exhaustion [2,3]. The risk of infection, the fear of contamination, social isolation, and uncertainty were associated with work-related stress and anxiety among frontline health professionals [2]. In addition, lack of information about the novel virus, unpreparedness about the use of personal protection equipment (PPE), and changing policies have increased the levels of stress and uncertainty for all nurses, even for those who did not care for COVID patients [4]. Nurses globally appeared to share similar feelings of fear, moral conflict, need for preparedness and safety, sense of duty, and exhaustion [5]. Issues of vulnerability, family protection from possible infection, and yet professional commitment and collegiality appeared to be issues of significant importance for nurses all over the world [6]. Moreover, the COVID-19 pandemic led to unpredictable changes in health care organizations and nursing practice. Nurses faced many challenges such as high and abrupt demands in pandemic-related care, barriers in communicating with patients and families, and strict boundaries in physical contact with their patients [7]. Excessive workload, role ambiguity, and interpersonal conflicts at work, seemed to further impact nursing practice and patient care and expand stress, anxiety, and depression among nurses [8]. 

In addition, health care systems were not prepared to tackle pandemic and this lack of readiness was manifested through constantly evolving guidelines that caused confusion and discomfort among nurses [9,10,11,12,13]. This led to non-compliance and care inconsistencies, while nurses’ performance in highly dependent clinical environments was hindered due to a lack of evidence-based treatment, insufficient knowledge about caring for COVID patients, poor patient prognosis, and lack of family presence [13,14,15,16,17]. 

Plenty of studies have been conducted worldwide to assess the psychological impact and the burnout frequency among frontline nurses during the pandemic [4,5,14,18,19,20]. Fewer focused on investigating the experiences of nurses working in general wards and the impact of the pandemic outbreak on the provision of nursing care [21,22,23].

In Greece, research evidence about the COVID-19 outbreak, has been mainly focused on investigating the mental health and the psychological impact of healthcare professionals and the general population, as well as the socio-economic effects and the health system responses to the pandemic crisis [24,25,26,27,28,29,30]. Data regarding nurses’ experience from the first wave of the pandemic is lacking as research in this specific area seems to be rather limited in Greece. 

In this respect, the aim of the present study was to describe how Greek nurses working in non-COVID units experienced the virus outbreak during the first wave of the pandemic. Furthermore, the impact of the pandemic on the provision of nursing care was explored.

Studying the nurses’ experiences from the first wave of the pandemic is considered important, as the COVID-19 outbreak was an unprecedented, extraordinary public health challenge that brought up the need for new adjustments and innovative interventions in the health care field. This was an epoch-making and yet critical for the life and health era that nurses encountered for the first time. In this era when societies and health care systems were challenged by an unexpected and life-threatening event, nurses undertook the most crucial role of controlling the pandemic and caring for their patients. The nurses’ experience at the very beginning of the pandemic, cannot be repeated or reoccur. 

For this reason, the evidence raised from this study is unique, and it can be used as a reference point on how health professionals and health systems adapt and evolve when they face such challenges. Although, after the first wave of the pandemic, more waves have followed, the data derived from the very first days of the pandemic outbreak can be considered exclusive knowledge that may globally impact the further development of nursing science and practice. 

## 2. Method

### 2.1. Design 

A descriptive qualitative research design was applied using a content analysis approach. Qualitative description designs are particularly relevant when little is known about a phenomenon and the researchers seek information directly from those experiencing the phenomenon under investigation [31,32]. 

As Bradshaw [32] states, qualitative description research lies within the naturalistic paradigm, which supports the researcher to gain an understanding of a phenomenon through accessing the meanings conveyed by the study participants. The inherent inductive process, the subjectivity of the experience of the participant and the researcher, the active participation of the researcher in the research process, and the collection of the data in the natural setting of the participants who experience the phenomenon, are some of the main philosophical underpinnings of qualitative description approach [32]. To a further extent, evidence from qualitative description studies may offer methodologically sound and straightforward guidance on how to enhance healthcare practice, to foster more attentive work on the subject under investigation, and enforce organizational leadership and policies [31,32]. 

The researchers in the present study aimed to provide a rich description of the participants’ experience and perspectives during the first days of the COVID-19 outbreak, a phenomenon for which little is known. Displaying an account of events and experiences as these were described from the participants’ viewpoint was considered essential for gaining in-depth knowledge directly from those who encountered the phenomenon under study. 

Learning from the participants’ descriptions and using this knowledge to guide policy making and reframe practice are two of the inherently valuable attributes of descriptive qualitative designs [31,32]. 

For this purpose, this particular design was considered the most suitable for presenting how nurses experienced the COVID-19 outbreak during the first wave of the pandemic, within their clinical context. 

### 2.2. Context and Participants

Registered nurses from the medical sector of a general hospital in Athens, Greece, participated in the study. A purposive sampling strategy was used to select study participants who provided nursing care in medical wards during the onset of the pandemic and had an adequate experience of the phenomenon under study. The inclusion criteria involved: (a) having a Bachelor’s degree in Nursing, and (b) having at least two years of working experience in the clinical sector, as this time period was considered essential for the study participants to thoroughly present and discuss their experience regarding the research topic [33]. Nurse assistants and other healthcare professionals were excluded.

One member of the research team contacted the nurse managers of the proposed study sites and informed them about the nature and the aim of the research, so to gain their support and assistance in conducting the study. Informal meetings were held with potential participants within the hospital. Through these meetings, the study was advertised and appropriate information regarding the context of the research was provided to the participants. This process enhanced subjects’ recruitment. Finally, ten female nurses from the medical sector participated and none of the participants withdrew from the study. The participants’ working experience ranged from 7 to 25 years. They all had a Bachelor’s degree in Nursing, while nine of them were MSc graduates. 

### 2.3. Data Collection

The data collection phase was carried out from February to June 2020. In-depth, semi-structured interviews were used for gathering detailed information about the nurses’ views and experiences. Interviews are considered the main tool of qualitative research, as it allows the researcher to gain a thorough understanding of the perceptions and beliefs of the people being asked [34]. The semi-structured interview consists of a flexible set of predefined questions and is used by the researcher as a guide to the issues which are considered important to cover within the interview [35]. It further provides the opportunity for the study participants to discuss and reveal their insights on complex or sensitive matters [36]. 

The interviews were conducted by a member of the authoring team who was experienced in applying qualitative interviewing techniques. The interviewer was a nurse, who was familiar with the context of the research without being actively involved in the work of the participants. This had a twofold effect, on the one hand, the participants felt comfortable expressing their opinions regarding the study topic, while, on the other hand, the appropriate neutrality regarding the role of the interviewer, was maintained [37]. Appropriate interviewing techniques were applied for encouraging the participants to provide detailed data [36]. The place and the time of interview were selected by the participants. Four interviews were conducted in a private area within the hospital and six interviews were arranged and completed via skype due to COVID-19 restrictions. 

In the frame of the interviews, study participants provided evidence of their feelings and experience of the pandemic outbreak. Open-ended questions are displayed in Table 1. 

The interviews lasted from 20 to 30 min. They were all recorded and verbatim transcription took place immediately after the completion of each interview. All interviews were conducted in Greek and a backward translation technique was applied when the text was translated to English. This technique ensured the accuracy of translation and the correct rendering of the extracts’ meaning [38]. Data saturation was achieved as no additional new information has been attained and further coding was not feasible after the ninth interview. Furthermore, saturation was also determined by the richness and thickness of the data which was obtained throughout the data collection phase. A wealth of the information gathered provided the best opportunity to answer the research question.

### 2.4. Data Analysis

Data were analyzed using a content analysis process. According to Sandelowski, this type of analysis fits better with the “straight description” of the data gathered in qualitative descriptive research designs [31]. Six steps of data analysis were used following Braun and Clarke framework [39], including familiarization with the data, coding the data, generating initial themes, reviewing and developing themes, refining, defining, and naming themes, and finally producing the report.

A member of the authoring team conducted the data analysis. In the first phase listening and reading the data repeatedly allows the analyst to be familiar with the data. Initial codes were formed (e.g., *emotional stress*) that identified basic elements of data (e.g., *uncertainty and fear*) having a meaning for the phenomenon under investigation. Following that, codes involving similar concepts were sorted out, leading to the generation of initial themes (e.g., *getting an emotional shock*). Reviewing and refining the initially formed themes led to the formulation of the final themes and subthemes accordingly, which reflected the essence of our data (Table 2).

### 2.5. Ethics

Before the commencement of the study ethical approval was requested and granted by the hospital’s Scientific Board. (Ref. No. 15/9-7-2019). Furthermore, an informed consent form was given and signed by each participant before data collection. The participants were fully informed about anonymity and confidentiality issues. Voluntary participation and the subjects’ right to withdraw from the study at any time without any penalty were also stressed. Permission was also given by the participants for tape recording the interview. To protect personal data code numbers were given to each one of the participants. Therefore, the interview excerpts used to illustrate the research findings, did not contain any identifying data. 

### 2.6. Credibility of Research 

Korstjens and Moser, suggest some techniques for enhancing trustworthiness in qualitative research [40]. Analyst triangulation and peer examination were two of the techniques applied in the present study. The former involved a second analyst who engaged to review the study findings and search for hidden concepts that might be overlooked. In this way, the integrity of the findings was reinforced. The latter involved rigorous feedback regarding the method the researchers’ personal involvement and possible bias throughout the data collection and data analysis phase. Member check was also applied for confirming the findings’ credibility [41,42]. The COREQ guidelines were used for reporting qualitative research as suggested by Tong et al. [43]. 

## 3. Findings 

Ten female University graduate nurses participated in the present study. They were all employed as staff nurses in the medical sector with a minimum overall working experience of seven years. The demographic characteristics of study participants are presented in Table 3. 

Data analysis revealed three themes namely, (a) emotional burden, (b) professional commitment, and (c) abrupt changes. Six subthemes were formulated and assimilated under each main theme respectively (Table 4). 

### 3.1. Emotional Burden

Fear and uncertainty were the predominant feelings that participants experienced during the pandemic outbreak. Emotional distress was derived and enhanced by the unprecedented feelings and the ambiguous changes that an unknown situation brought up. 

#### 3.1.1. The Fear of Contamination 

The nurses’ anxiety about being contaminated was communicated throughout the interviews. The fear of being contaminated by the virus and transmitting it to their beloved ones was clearly reported. This feeling appeared to dominate the nurses’ thoughts, despite the fact that in some cases remained hidden. 


*Mainly it was fear…. fear of not being contaminated… we were more afraid for our families not to be infected rather than for us. Generally, there was fear inside us but we tried to hide it, not to be shown to others (p1).*



*We were afraid of being contaminated, because potentially everyone could be positive (p2).*



*I was very worried… I wanted to stay safe and healthy, I was afraid of bringing the virus to my family (p9).*


#### 3.1.2. The Uncertainty 

Feelings of uncertainty and vulnerability were also reported by the study participants. These feelings were reinforced by the COVID-19 restrictive measures, such as quarantine and lockdown. 


*It was a strange situation, there was a strange silence in the night shifts, it was uncertainty… and fear (p6).*



*There was fear and uncertainty… because at a time when everyone was telling people to stay at home we had to go out and come to the hospital…(p4).*


This ambiguity seemed also to impact the nurses’ perceived capability to provide the best patient care. 


*The situation is very ambiguous…The fear is still here, I think it will be with us for a long time. You want to do the best for the patient but on the other hand, you feel unprotected (p5).*


### 3.2. Professional Commitment 

The nurses’ unique professional role was highlighted throughout the participants’ statements, as the respondents were focused on their professional duty and the need to be next to the patients. 

#### 3.2.1. Caring for Patients’ Well-Being

Many of the participants appeared to be frustrated or even reluctant to go to work in an era during which measures of social distance and lockdowns were applied worldwide. However, being next to the patients and provide care to them prevailed over hesitation and anxiety. 


*Our job is to be next to the patient, you have to take care for the sick person you wear your mask and gloves and you are there … (p3).*



*I was frustrated… I didn’t want to go to work, I refused to go…, on the other hand, I knew that the patients were there, waiting for me… (p5).*



*We were anxious… you are not sure if a patient has the virus or not… then you start working and after a while, you forget the virus…You cannot leave the patient… you cannot ignore him… (p6).*


#### 3.2.2. The Work Does Not Stop

The nurses continued to go to work as they used to do before the pandemic, since that was an imperative duty for them. PPE was referred to as an additional accessory, involved in patient care, necessary though for safely continuing their work. 


*We continue to work as we did before. Our work, our care did not stop… we work as we used to do (laughing) having some additional accessories… the mask…the gloves (p1).*



*The work did not stop, we pay more attention to personal protection measures just to protect ourselves and our loved ones (p3).*


Nurses also stressed their commitment to their work despite their psychological distress. 


*I am psychologically affected … but I am still doing what I did before. When I have to go to the patients I’ll go. When I have to give care, I’ll do it. Nursing cannot stop (p8).*


### 3.3. Abrupt Changes 

During the first wave of the COVID-19 pandemic, unexpected and massive changes rapidly evolved worldwide. The health care systems were severely affected and major organizational alterations were implemented. This rapidly changing environment impacted both the healthcare professionals and the care provided within the healthcare organizations. 

#### 3.3.1. The Organizational Shift

Immediate and abrupt changes at an organizational level created frustration and anxiety for nurses. These new demands called for rapid adjustments that nurses tried to incorporate into their daily work. 


*Things were changed over a night…the guidelines changed every day, many changes happened at an organizational level, reorganization of departments, staff deployment…many changes were very annoying to us, to our work (p2).*



*At the end of the day, I do not know what bothered me the most, the fear of the virus or the changes that the top management imposed to us, the lack of information, the lack of equipment and material…(p5).*



*There were big changes in the hospital, many new things that worried me, new rules, new equipment, new protection measures… I tried to adjust and as time goes by I got used more or less to all these… (p4).*


#### 3.3.2. The Impact on Nurses and Care

Within this rapidly changing environment, both organizational and personal strains appeared to affect nurses’ performance in the delivery of care. Some of them reported that the use of PPE and the psychological burden caused certain limitations in dealing with the patient. 


*The pandemic affected me… the patient was alone and I could not go in and talk to him as I wanted to because of the protective measures… that was horrible (p10).*



*I have changed a lot, I am more distant with the patients, I am more distant at home, everything changed… the way you approach the patients, the things you do when you go back home, everything is new… and I am not relaxed the way I was before (p3).*



*Nobody remained intact in this situation…. although the work was reduced in our ward, which is a good thing, we were not happy…we were affected psychologically and this impacted on our work (p7).*


Despite, the difficulties that COVID-19 caused, some nurses pointed out some positive aspects of the care delivered during the pandemic era. As previously reported, the better nurse/patient ratio due to reduced admissions and the strict compliance to safety measures, supported improved care. Feelings of anxiety and stress remained but this seemed not to hinder the provision of quality nursing care. 


*The work in our ward was better, we had fewer patients… we were not physically tired but we were more stressed…I suppose that now we have fewer patients and we can provide better care to them (p3).*



*We became more focused, more cautious, more diligent, when approaching patients, when we use the protection equipment, when we do hand hygiene (p9).*



*The pandemic added some more stress on me, but COVID did not stop me from providing excellent nursing care… (p8).*


## 4. Discussion

The findings of the present study emphasized the nurses’ experiences during the first wave COVID-19 pandemic. Emotional burden and feelings of fear, anxiety, and uncertainty, appeared to have a crucial effect on nurses’ life and the care provided. However, the professional commitment and the nurses’ effort to provide excellent nursing care remained high. In a rapidly changing environment involving a variety of stressors and burdens, nurses managed to show empathy for the patient, thoroughness, responsibility, and professional commitment. They faced abrupt changes including staff reallocation, rotation and understaffing, continuous use of PPE, and novel guidelines which should be strictly followed. These changes created a frustrating situation that impacted both nurses and patient care. Within this shifting and uncertain environment, some positive aspects were pointed out. These were mainly related to the reduced workload for non-COVID clinics, caused by the decreased number of patient admissions for routine treatment procedures. This enabled nurses to focus more intensively on patients’ needs and safety and to provide thus a better quality of nursing care. 

### 4.1. Emotional Burden: Fear and Uncertainty

Our findings revealed that fear and uncertainty were the most prevalent emotions among nurses throughout the pandemic’s spread. The unexpected sentiments and the abrupt changes that COVID-19 brought up, resulted in emotional distress. The nurses’ fear of becoming infected with the virus and infecting their loved ones was plainly expressed throughout the interviews. 

Stress, anxiety, and fear were reported in the relevant literature as the most common emotions caused by the COVID-19 pandemic [4,5,6,9,13,44,45]. The fear of being contaminated was demonstrated specifically by the front-line nurses as they were in close contact with COVID-19 patients. Fear of death was also reported and several studies underscore that infection or death of colleagues and other health care professionals exacerbated nurses’ fears and anxiety [12,16,17,45,46,47,48].

In line with our findings, the potential risk of transmitting the virus to their family and loved ones was also highlighted in the relevant literature as a factor that increased nurses’ anxiety [11,49]. Emotional strain due to lack of understanding and feelings of guilt or self-blame for the infection of family members have been also reported by the frontline nurses [13,17,50]. 

Participants in the present study also expressed feelings of uncertainty and vulnerability that were reinforced by the COVID-19 restrictive measures and lockdown and influenced the nurses’ ability to provide the best care to the patients. 

In the same vein, findings from relevant studies have reported that nurses at the beginning of the pandemic have been left with a sense of uncertainty. As Galehdar et al., and Góes et al., point out the disease was unprecedented and therefore nurses worked under pressure due to a lack of available scientific information [11,45]. This ambiguous and unpredictable situation has inevitably brought about feelings of fear, uncertainty, and vulnerability in front-line nurses [47,51]. Uncertainty was also related to the variety and content of the information disseminated via media, which was referred to as one of the most stressful factors influencing nurses’ emotions [4,10,11,15,44,45,49,52]. In addition, as the pandemic spread, nurses reported concerns about their future professional and personal lives [52], and this uncertainty and emotional stress led to physical and mental exhaustion, hindering thus their professional performance [11,16,47,48,49,50,53]. 

### 4.2. Professional Commitment: The Working Environment and the Patient

The importance of the nurses’ distinct professional role was emphasized in the present study throughout the nurses’ responses. The study participants were focused on their professional responsibilities and clearly demonstrated the necessity to care for their patients’ well-being.

The significance of professional values and commitment were emphasized in various studies which explored frontline nurses’ attitudes during the pandemic [9,11,12,45,49,50,54]. Specifically, a professional commitment was the foundation of nurses’ motivation, while professional values and responsibility supported their commitment to work on the frontline without any hesitation and despite the fear and the risk of contamination [52]. Similarly, to the findings of the present study, Galehdar et al., Fan et al., and Góes et al., pointed out that nurses during the pandemic have made great efforts to preserve patients’ well-being and to assist their family [11,45,50]. As such, opportunities to achieve their professional goals and values which enhanced their professional role and identity were raised [6,11,16,17,47,50]. 

Our study participants displayed their determination to keep working and caring for the patient as they did before the pandemic. PPE was regarded as an additional safety measure, which helped nurses to respond to their working responsibilities and to protect themselves and their families from the virus. This finding was in contrast with evidence raised from similar research in which nurses reported that PPE impeded vision and communication with the patient and prevented the provision of appropriate care [12,17,51,52,53]. Despite that, nurses maintained their professional commitment and expressed a sense of pride in achieving their professional goals [9,11,52]. 

### 4.3. Organizational Changes: The Impact on Nurses and Patient Care

The findings of the present study highlighted the unexpected changes that occurred during the first wave of the COVID-19 pandemic. Nurse participants referred to the rapidly changing environment that required immediate adjustments at an organizational, professional, and personal level. The nurses who tried to adapt to the new conditions felt frustration and anxiety by these sudden changes. 

In the same vein, Fernadez et al. [15] illustrated the pressure on the nursing workforce during the COVID-19 outbreak which urged nurses to adapt to changes quickly, in a rather difficult and highly demanding environment. Bambi et al. [55] commented on the sudden organizational changes that were imposed by the hospital managers in order to effectively cope with the pandemic crisis, by pointing out that some of them (e.g., inappropriate staff skill mix due to staff shortage) hindered patients’ safety and care. Furthermore, relevant research evidence confirmed the nurses’ physical and psychological stress caused by the unexpected life and work changes. In this respect, nurses referred to supportive relationships, care, and understanding from colleagues and upper management as factors relieving negative feelings and facilitating a successful transition to new work patterns [5,12,15,17,52]. 

Our findings highlighted also the nurses’ personal constraints, and the impediments to caring practices and communication caused by COVID-19. Thrysoee et al. [56] found that social distancing and fear of contamination affected nurses’ life and caring practices. 

In addition, the wide-ranging changes in health care organizations and the assignment of unfamiliar tasks to the staff increased nurses’ burnout levels. This impacted negatively on nurses’ personal and professional performance [57,58]. 

Despite the challenges faced, the nurses in our study identified a few positive aspects that supported the provision of care during the pandemic era. For example, improved nurse/patient ratio, lower number of admissions, and compliance with policies and safety procedures were referred to as factors that improved the standard of care provided in their wards. 

In line with the findings of the present study, von Vogelsang et al. [59], who investigated the deficiencies in nursing care during the COVID-19 pandemic at inpatient wards, highlighted the crucial role of lower patients’ admissions and the maintenance of nurse/patient ratio. However, during the first wave of the COVID-19 pandemic, the standards of care at non-COVID wards were regarded to be inadequate [60]. Even more, the pandemic crisis raised issues of inconsistencies in nursing care and low nursing performance due to nurses’ mental health symptoms [61,62]. Stress, emotional exhaustion, and anxiety, impaired nurses’ clinical performance and their ability to successfully attain the required nursing tasks [63,64].

Nurses in the present study stated that although levels of stress have increased due to the pandemic, this did not prevent them from providing excellent nursing care. This discrepancy between our findings and those of other studies might be explained by the nurses’ strong sense of professional commitment and responsibility toward the patients. Furthermore, it might be stated that nurses in general wards experienced the pandemic outbreak less intensively [15,58,60,65]. Supporting health professionals who work both on the frontline and in the general wards is of critical importance and the role of nursing management in developing supporting strategies and maintaining quality practice has been widely recognized [5,66,67]. 

## 5. Limitations

This study involved ten nurses working in the medical sector during the first wave of the COVID-19 outbreak. Since this condition may limit the applicability of our findings, further research on this topic is recommended involving more participants from a variety of clinical sectors. Nurses’ views of working in peripheral health care organizations located in non-metropolitan cities, may also contribute to further development of the knowledge gained so far in the relevant research field. Conducting interviews using alternative interview modes such as skype can also limit the researcher’s access to body language and may inhibit the communication employed in face-to-face interviews. As such, the findings of this study should be viewed under these limitations. 

## 6. Conclusions 

Our findings stated that emotional disruption, fear, uncertainty, and frustration were amongst the dominant feelings that nurses working in general wards experienced. The abrupt changes caused by the pandemic drastically impacted nurses and care as the participants revealed that nobody remained intact. Professional values and commitment prevailed through over adversities and the virus did not stop nurses from providing excellent care despite the pandemic burdens. Within this changing environment, nurses identified some positive aspects and it appears that the pandemic created opportunities for practice improvement through increased caution, thoroughness, and accuracy in nursing care. This is an unexpected finding, as the prevailing perception at the period was that Greek nurses were under tremendous stress and overwhelmed due to the strain imposed on the health care system. This novel to us evidence may be utilized to enrich the current scientific knowledge on this topic and inform practice by providing an in-depth understanding of how nurses react and adapt in unexpected crisis situations. Further research on the experiences of health managers and leaders regarding organizational and governmental changes imposed by the pandemic, may also contribute to successful policy-making interventions during health crises.

## Figures and Tables

**Table 1 healthcare-10-01406-t001:** Interview scheme.

Interview Questions
How did you experience the COVID-19 outbreak at work? How this event impacted care How this event impacted you How did you feel in terms of providing nursing care during that period? Which were the bad and the good parts of it How did you feel about the new conditions that evolved in your work? Other comments, thoughts, and feelings you would like to add

**Table 2 healthcare-10-01406-t002:** Coding and Generation of final themes.

Units of Analysis (Key Words and Phrases)	Basic Elements of Data	Initial Codes	Initially Generated Themes	Final Themes
“*there was fear inside us*” “*it was fear and uncertainty*”	Uncertainty and fear	Emotional stress	Getting an emotional shock	Emotional Burden
“*the patients were there*” “*the work did not stop*”	The patient and my work	Sense of duty	Being a responsible professional	Professional Commitment
“*many changes happened at an organizational level*” “*nobody remained intact*”	Shifts in personal and professional life	Reformed life	Experiencing a complete transformation	Abrupt Changes

**Table 3 healthcare-10-01406-t003:** Demographic characteristics of study participants.

Participants’ Pseudonyms	Age	Marital Status	Years of Working Experience	Education
p1	47	married	24	BSc, MSc
p2	38	single	12	BSc, MSc
p3	47	married	25	BSc, MSc
p4	35	single	15	BSc, MSc
p5	31	single	8	BSc, MSc
p6	47	single	25	BSc, MSc
p7	37	married	13	BSc, MSc
p8	39	married	14	BSc, MSc
p9	42	single	17	BSc, MSc
p10	30	single	7	BSc

**Table 4 healthcare-10-01406-t004:** Themes and subthemes.

Themes	Subthemes
A. Emotional Burden	
	A1. The Fear of Contamination
	A2. The Uncertainty
B. Professional Commitment	
	B1. Caring for patients’ well-being
	B2. The Work does not stop
C. Abrupt Changes	
	C1. The Organizational Shift
	C2. The Impact on Nurses and Care

## Data Availability

Data generated during the present study is not possible to be shared due to issues of subjects’ privacy and confidentiality.

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
