# Peer review of "More by Authors Links"

_healthcare, 2022, doi:10.3390/healthcare10081406_

Round 1
Reviewer 1 Report
Dear authors,
Thank you for the opportunity to review the manuscript. The study is well-written and has revealed certain interesting insights into the experiences of nurses in Greece regarding their care in the non-COVID-19 wards during COVID-19. Furthermore, this study has merits in its contribution to knowledge about the experiences as the authors have made good attempts to compare experiences between nurses working in COVID-19 and non-covid-19 wards.
I have some comments for the authors to consider for enhancing their manuscript:
L126: The first part of the sentence requires some rephrasing.
L133: I am unsure of the number and types of questions in the interview guide? Can the authors append the interview guide for readers’ reference?
L414 & L357: “recognised” and “organisational”– suggested to be consistent with the use of either American or British English, as there is a mixed-use of both.
Results: Violent changes. May I ask why the authors chose the word “Violent changes” for this theme? I have tried to see if this word is reflected or mentioned in the subthemes and quotes but did not see it. The word “violence” might confuse readers who think that some sort of aggressive event has happened. Or are the authors trying to mean “volatile changes”? Can the authors revisit this theme just to be sure that the theme is correctly worded?
Others: There is no mention of data saturation in the manuscript. This issue needs to be addressed in the limitation if data saturation was not determined as required for the “trustworthiness” of qualitative research.
Others (2): Some inconsistent font and size are being observed in the manuscript. I trust that the editors will work with the authors to rectify this aspect.
Reviewer 2 Report
The manuscript presented by the authors has an important theme, although it is true, that talking about the first wave of covid is already a bit obsolete.
The introduction needs to incorporate more relevant literature regarding the topic to be addressed in different European environments and outside it.
The goal is tremendously generalized, Greek nurses are something unattainable and 0 extrapolable, the population needs to be specified.
An intentional sampling strategy was used, nothing else is specified. Regarding the inclusion criteria, you should specify something else, what you propose is only a criterion.
It does not refer to the principle of saturation of the discourse, it simply talks about 10 interviews.
The results are not rigorously exposed, we do not know on the basis of which theory has been analyzed, we do not know the initial categories and which have been the emerging ones...
It needs to improve the many methodological fissures that the manuscript has.
It is not currently suitable for publication.
Reviewer 3 Report
1.The title does not reflect the true objective of the research, which is not the quality of nursing care, but rather the description of how certain nursing staff spent the COVID-19 pandemic. Therefore, I think they should change the title and adjust it to the real objective of the study.
2.Throughout the text, I believe that it confuses the obligatory professional care that must be dispensed by nurses with quality nursing care. This happens because they do not define what they mean by quality in nursing care (an aspect that does not appear in the introduction). From my point of view, when we analyse the quality of care, we must contrast it with other data taken previously or with previously established standards, as well as collecting data from the people on whom the nursing care and attention falls. To say that there has been quality in nursing care with the measures that have been collected and the subjects included in the research is, in my opinion, very adventurous and rather subjective.
3.I believe that the introduction does not go into the topics or variables that are then highlighted in the research in the results section. It should include the quality of nursing care, the results obtained in other research on fears of contagion, the stress caused by the COVID-19 situation, among others.
4.From my point of view, the objective should go at the end of the introduction or justification and not in the method.
5.The method section is very confusing. I consider that the information should be structured following the classic sections, participants, instruments and procedure.
6.When you talk about the participants, you should indicate the number of participants and whether or not they are representative of the sample, at least of the hospital. You should also clearly specify the sex of the participants although it may be obvious.
7.When talking about the instrument used, even if it is a semi-structured interview, you should include some sample questions or indicate what the questions were about.
8.The procedure should be described more clearly and accurately.
9.It is not clear who did the categorisation of the contents, whether it was arrived at by inter-judge agreement, etc. The same applies to the coding of the contents.
10.In the results section, even if the research is qualitative, quantitative data can always be provided to clarify, for example, how many arguments were included in the different coding categories.
11.I think that with the results presented in the study it is very daring to discuss them, for me, it is a question of whether you believe it or not. I have no data to be able to judge whether or not the results obtained are in line with other research included in the paper.
1.The title does not reflect the true objective of the research, which is not the quality of nursing care, but rather the description of how certain nursing staff spent the COVID-19 pandemic. Therefore, I think they should change the title and adjust it to the real objective of the study.
2.Throughout the text, I believe that it confuses the obligatory professional care that must be dispensed by nurses with quality nursing care. This happens because they do not define what they mean by quality in nursing care (an aspect that does not appear in the introduction). From my point of view, when we analyse the quality of care, we must contrast it with other data taken previously or with previously established standards, as well as collecting data from the people on whom the nursing care and attention falls. To say that there has been quality in nursing care with the measures that have been collected and the subjects included in the research is, in my opinion, very adventurous and rather subjective.
3.I believe that the introduction does not go into the topics or variables that are then highlighted in the research in the results section. It should include the quality of nursing care, the results obtained in other research on fears of contagion, the stress caused by the COVID-19 situation, among others.
4.From my point of view, the objective should go at the end of the introduction or justification and not in the method.
5.The method section is very confusing. I consider that the information should be structured following the classic sections, participants, instruments and procedure.
6.When you talk about the participants, you should indicate the number of participants and whether or not they are representative of the sample, at least of the hospital. You should also clearly specify the sex of the participants although it may be obvious.
7.When talking about the instrument used, even if it is a semi-structured interview, you should include some sample questions or indicate what the questions were about.
8.The procedure should be described more clearly and accurately.
9.It is not clear who did the categorisation of the contents, whether it was arrived at by inter-judge agreement, etc. The same applies to the coding of the contents.
10.In the results section, even if the research is qualitative, quantitative data can always be provided to clarify, for example, how many arguments were included in the different coding categories.
11.I think that with the results presented in the study it is very daring to discuss them, for me, it is a question of whether you believe it or not. I have no data to be able to judge whether or not the results obtained are in line with other research included in the paper.
1.The title does not reflect the true objective of the research, which is not the quality of nursing care, but rather the description of how certain nursing staff spent the COVID-19 pandemic. Therefore, I think they should change the title and adjust it to the real objective of the study.
2.Throughout the text, I believe that it confuses the obligatory professional care that must be dispensed by nurses with quality nursing care. This happens because they do not define what they mean by quality in nursing care (an aspect that does not appear in the introduction). From my point of view, when we analyse the quality of care, we must contrast it with other data taken previously or with previously established standards, as well as collecting data from the people on whom the nursing care and attention falls. To say that there has been quality in nursing care with the measures that have been collected and the subjects included in the research is, in my opinion, very adventurous and rather subjective.
3.I believe that the introduction does not go into the topics or variables that are then highlighted in the research in the results section. It should include the quality of nursing care, the results obtained in other research on fears of contagion, the stress caused by the COVID-19 situation, among others.
4.From my point of view, the objective should go at the end of the introduction or justification and not in the method.
5.The method section is very confusing. I consider that the information should be structured following the classic sections, participants, instruments and procedure.
6.When you talk about the participants, you should indicate the number of participants and whether or not they are representative of the sample, at least of the hospital. You should also clearly specify the sex of the participants although it may be obvious.
7.When talking about the instrument used, even if it is a semi-structured interview, you should include some sample questions or indicate what the questions were about.
8.The procedure should be described more clearly and accurately.
9.It is not clear who did the categorisation of the contents, whether it was arrived at by inter-judge agreement, etc. The same applies to the coding of the contents.
10.In the results section, even if the research is qualitative, quantitative data can always be provided to clarify, for example, how many arguments were included in the different coding categories.
11.I think that with the results presented in the study it is very daring to discuss them, for me, it is a question of whether you believe it or not. I have no data to be able to judge whether or not the results obtained are in line with other research included in the paper.
1.The title does not reflect the true objective of the research, which is not the quality of nursing care, but rather the description of how certain nursing staff spent the COVID-19 pandemic. Therefore, I think they should change the title and adjust it to the real objective of the study.
2.Throughout the text, I believe that it confuses the obligatory professional care that must be dispensed by nurses with quality nursing care. This happens because they do not define what they mean by quality in nursing care (an aspect that does not appear in the introduction). From my point of view, when we analyse the quality of care, we must contrast it with other data taken previously or with previously established standards, as well as collecting data from the people on whom the nursing care and attention falls. To say that there has been quality in nursing care with the measures that have been collected and the subjects included in the research is, in my opinion, very adventurous and rather subjective.
3.I believe that the introduction does not go into the topics or variables that are then highlighted in the research in the results section. It should include the quality of nursing care, the results obtained in other research on fears of contagion, the stress caused by the COVID-19 situation, among others.
4.From my point of view, the objective should go at the end of the introduction or justification and not in the method.
5.The method section is very confusing. I consider that the information should be structured following the classic sections, participants, instruments and procedure.
6.When you talk about the participants, you should indicate the number of participants and whether or not they are representative of the sample, at least of the hospital. You should also clearly specify the sex of the participants although it may be obvious.
7.When talking about the instrument used, even if it is a semi-structured interview, you should include some sample questions or indicate what the questions were about.
8.The procedure should be described more clearly and accurately.
9.It is not clear who did the categorisation of the contents, whether it was arrived at by inter-judge agreement, etc. The same applies to the coding of the contents.
10.In the results section, even if the research is qualitative, quantitative data can always be provided to clarify, for example, how many arguments were included in the different coding categories.
11.I think that with the results presented in the study it is very daring to discuss them, for me, it is a question of whether you believe it or not. I have no data to be able to judge whether or not the results obtained are in line with other research included in the paper.
1.The title does not reflect the true objective of the research, which is not the quality of nursing care, but rather the description of how certain nursing staff spent the COVID-19 pandemic. Therefore, I think they should change the title and adjust it to the real objective of the study.
2.Throughout the text, I believe that it confuses the obligatory professional care that must be dispensed by nurses with quality nursing care. This happens because they do not define what they mean by quality in nursing care (an aspect that does not appear in the introduction). From my point of view, when we analyse the quality of care, we must contrast it with other data taken previously or with previously established standards, as well as collecting data from the people on whom the nursing care and attention falls. To say that there has been quality in nursing care with the measures that have been collected and the subjects included in the research is, in my opinion, very adventurous and rather subjective.
3.I believe that the introduction does not go into the topics or variables that are then highlighted in the research in the results section. It should include the quality of nursing care, the results obtained in other research on fears of contagion, the stress caused by the COVID-19 situation, among others.
4.From my point of view, the objective should go at the end of the introduction or justification and not in the method.
5.The method section is very confusing. I consider that the information should be structured following the classic sections, participants, instruments and procedure.
6.When you talk about the participants, you should indicate the number of participants and whether or not they are representative of the sample, at least of the hospital. You should also clearly specify the sex of the participants although it may be obvious.
7.When talking about the instrument used, even if it is a semi-structured interview, you should include some sample questions or indicate what the questions were about.
8.The procedure should be described more clearly and accurately.
9.It is not clear who did the categorisation of the contents, whether it was arrived at by inter-judge agreement, etc. The same applies to the coding of the contents.
10.In the results section, even if the research is qualitative, quantitative data can always be provided to clarify, for example, how many arguments were included in the different coding categories.
11.I think that with the results presented in the study it is very daring to discuss them, for me, it is a question of whether you believe it or not. I have no data to be able to judge whether or not the results obtained are in line with other research included in the paper.
1.The title does not reflect the true objective of the research, which is not the quality of nursing care, but rather the description of how certain nursing staff spent the COVID-19 pandemic. Therefore, I think they should change the title and adjust it to the real objective of the study.
2.Throughout the text, I believe that it confuses the obligatory professional care that must be dispensed by nurses with quality nursing care. This happens because they do not define what they mean by quality in nursing care (an aspect that does not appear in the introduction). From my point of view, when we analyse the quality of care, we must contrast it with other data taken previously or with previously established standards, as well as collecting data from the people on whom the nursing care and attention falls. To say that there has been quality in nursing care with the measures that have been collected and the subjects included in the research is, in my opinion, very adventurous and rather subjective.
3.I believe that the introduction does not go into the topics or variables that are then highlighted in the research in the results section. It should include the quality of nursing care, the results obtained in other research on fears of contagion, the stress caused by the COVID-19 situation, among others.
4.From my point of view, the objective should go at the end of the introduction or justification and not in the method.
5.The method section is very confusing. I consider that the information should be structured following the classic sections, participants, instruments and procedure.
6.When you talk about the participants, you should indicate the number of participants and whether or not they are representative of the sample, at least of the hospital. You should also clearly specify the sex of the participants although it may be obvious.
7.When talking about the instrument used, even if it is a semi-structured interview, you should include some sample questions or indicate what the questions were about.
8.The procedure should be described more clearly and accurately.
9.It is not clear who did the categorisation of the contents, whether it was arrived at by inter-judge agreement, etc. The same applies to the coding of the contents.
10.In the results section, even if the research is qualitative, quantitative data can always be provided to clarify, for example, how many arguments were included in the different coding categories.
11.I think that with the results presented in the study it is very daring to discuss them, for me, it is a question of whether you believe it or not. I have no data to be able to judge whether or not the results obtained are in line with other research included in the paper.
1.The title does not reflect the true objective of the research, which is not the quality of nursing care, but rather the description of how certain nursing staff spent the COVID-19 pandemic. Therefore, I think they should change the title and adjust it to the real objective of the study.
2.Throughout the text, I believe that it confuses the obligatory professional care that must be dispensed by nurses with quality nursing care. This happens because they do not define what they mean by quality in nursing care (an aspect that does not appear in the introduction). From my point of view, when we analyse the quality of care, we must contrast it with other data taken previously or with previously established standards, as well as collecting data from the people on whom the nursing care and attention falls. To say that there has been quality in nursing care with the measures that have been collected and the subjects included in the research is, in my opinion, very adventurous and rather subjective.
3.I believe that the introduction does not go into the topics or variables that are then highlighted in the research in the results section. It should include the quality of nursing care, the results obtained in other research on fears of contagion, the stress caused by the COVID-19 situation, among others.
4.From my point of view, the objective should go at the end of the introduction or justification and not in the method.
5.The method section is very confusing. I consider that the information should be structured following the classic sections, participants, instruments and procedure.
6.When you talk about the participants, you should indicate the number of participants and whether or not they are representative of the sample, at least of the hospital. You should also clearly specify the sex of the participants although it may be obvious.
7.When talking about the instrument used, even if it is a semi-structured interview, you should include some sample questions or indicate what the questions were about.
8.The procedure should be described more clearly and accurately.
9.It is not clear who did the categorisation of the contents, whether it was arrived at by inter-judge agreement, etc. The same applies to the coding of the contents.
10.In the results section, even if the research is qualitative, quantitative data can always be provided to clarify, for example, how many arguments were included in the different coding categories.
11.I think that with the results presented in the study it is very daring to discuss them, for me, it is a question of whether you believe it or not. I have no data to be able to judge whether or not the results obtained are in line with other research included in the paper.
1.The title does not reflect the true objective of the research, which is not the quality of nursing care, but rather the description of how certain nursing staff spent the COVID-19 pandemic. Therefore, I think they should change the title and adjust it to the real objective of the study.
2.Throughout the text, I believe that it confuses the obligatory professional care that must be dispensed by nurses with quality nursing care. This happens because they do not define what they mean by quality in nursing care (an aspect that does not appear in the introduction). From my point of view, when we analyse the quality of care, we must contrast it with other data taken previously or with previously established standards, as well as collecting data from the people on whom the nursing care and attention falls. To say that there has been quality in nursing care with the measures that have been collected and the subjects included in the research is, in my opinion, very adventurous and rather subjective.
3.I believe that the introduction does not go into the topics or variables that are then highlighted in the research in the results section. It should include the quality of nursing care, the results obtained in other research on fears of contagion, the stress caused by the COVID-19 situation, among others.
4.From my point of view, the objective should go at the end of the introduction or justification and not in the method.
5.The method section is very confusing. I consider that the information should be structured following the classic sections, participants, instruments and procedure.
6.When you talk about the participants, you should indicate the number of participants and whether or not they are representative of the sample, at least of the hospital. You should also clearly specify the sex of the participants although it may be obvious.
7.When talking about the instrument used, even if it is a semi-structured interview, you should include some sample questions or indicate what the questions were about.
8.The procedure should be described more clearly and accurately.
9.It is not clear who did the categorisation of the contents, whether it was arrived at by inter-judge agreement, etc. The same applies to the coding of the contents.
10.In the results section, even if the research is qualitative, quantitative data can always be provided to clarify, for example, how many arguments were included in the different coding categories.
11.I think that with the results presented in the study it is very daring to discuss them, for me, it is a question of whether you believe it or not. I have no data to be able to judge whether or not the results obtained are in line with other research included in the paper.
1.The title does not reflect the true objective of the research, which is not the quality of nursing care, but rather the description of how certain nursing staff spent the COVID-19 pandemic. Therefore, I think they should change the title and adjust it to the real objective of the study.
2.Throughout the text, I believe that it confuses the obligatory professional care that must be dispensed by nurses with quality nursing care. This happens because they do not define what they mean by quality in nursing care (an aspect that does not appear in the introduction). From my point of view, when we analyse the quality of care, we must contrast it with other data taken previously or with previously established standards, as well as collecting data from the people on whom the nursing care and attention falls. To say that there has been quality in nursing care with the measures that have been collected and the subjects included in the research is, in my opinion, very adventurous and rather subjective.
3.I believe that the introduction does not go into the topics or variables that are then highlighted in the research in the results section. It should include the quality of nursing care, the results obtained in other research on fears of contagion, the stress caused by the COVID-19 situation, among others.
4.From my point of view, the objective should go at the end of the introduction or justification and not in the method.
5.The method section is very confusing. I consider that the information should be structured following the classic sections, participants, instruments and procedure.
6.When you talk about the participants, you should indicate the number of participants and whether or not they are representative of the sample, at least of the hospital. You should also clearly specify the sex of the participants although it may be obvious.
7.When talking about the instrument used, even if it is a semi-structured interview, you should include some sample questions or indicate what the questions were about.
8.The procedure should be described more clearly and accurately.
9.It is not clear who did the categorisation of the contents, whether it was arrived at by inter-judge agreement, etc. The same applies to the coding of the contents.
10.In the results section, even if the research is qualitative, quantitative data can always be provided to clarify, for example, how many arguments were included in the different coding categories.
11.I think that with the results presented in the study it is very daring to discuss them, for me, it is a question of whether you believe it or not. I have no data to be able to judge whether or not the results obtained are in line with other research included in the paper.
1.The title does not reflect the true objective of the research, which is not the quality of nursing care, but rather the description of how certain nursing staff spent the COVID-19 pandemic. Therefore, I think they should change the title and adjust it to the real objective of the study.
2.Throughout the text, I believe that it confuses the obligatory professional care that must be dispensed by nurses with quality nursing care. This happens because they do not define what they mean by quality in nursing care (an aspect that does not appear in the introduction). From my point of view, when we analyse the quality of care, we must contrast it with other data taken previously or with previously established standards, as well as collecting data from the people on whom the nursing care and attention falls. To say that there has been quality in nursing care with the measures that have been collected and the subjects included in the research is, in my opinion, very adventurous and rather subjective.
3.I believe that the introduction does not go into the topics or variables that are then highlighted in the research in the results section. It should include the quality of nursing care, the results obtained in other research on fears of contagion, the stress caused by the COVID-19 situation, among others.
4.From my point of view, the objective should go at the end of the introduction or justification and not in the method.
5.The method section is very confusing. I consider that the information should be structured following the classic sections, participants, instruments and procedure.
6.When you talk about the participants, you should indicate the number of participants and whether or not they are representative of the sample, at least of the hospital. You should also clearly specify the sex of the participants although it may be obvious.
7.When talking about the instrument used, even if it is a semi-structured interview, you should include some sample questions or indicate what the questions were about.
8.The procedure should be described more clearly and accurately.
9.It is not clear who did the categorisation of the contents, whether it was arrived at by inter-judge agreement, etc. The same applies to the coding of the contents.
10.In the results section, even if the research is qualitative, quantitative data can always be provided to clarify, for example, how many arguments were included in the different coding categories.
11.I think that with the results presented in the study it is very daring to discuss them, for me, it is a question of whether you believe it or not. I have no data to be able to judge whether or not the results obtained are in line with other research included in the paper.
1.The title does not reflect the true objective of the research, which is not the quality of nursing care, but rather the description of how certain nursing staff spent the COVID-19 pandemic. Therefore, I think they should change the title and adjust it to the real objective of the study.
2.Throughout the text, I believe that it confuses the obligatory professional care that must be dispensed by nurses with quality nursing care. This happens because they do not define what they mean by quality in nursing care (an aspect that does not appear in the introduction). From my point of view, when we analyse the quality of care, we must contrast it with other data taken previously or with previously established standards, as well as collecting data from the people on whom the nursing care and attention falls. To say that there has been quality in nursing care with the measures that have been collected and the subjects included in the research is, in my opinion, very adventurous and rather subjective.
3.I believe that the introduction does not go into the topics or variables that are then highlighted in the research in the results section. It should include the quality of nursing care, the results obtained in other research on fears of contagion, the stress caused by the COVID-19 situation, among others.
4.From my point of view, the objective should go at the end of the introduction or justification and not in the method.
5.The method section is very confusing. I consider that the information should be structured following the classic sections, participants, instruments and procedure.
6.When you talk about the participants, you should indicate the number of participants and whether or not they are representative of the sample, at least of the hospital. You should also clearly specify the sex of the participants although it may be obvious.
7.When talking about the instrument used, even if it is a semi-structured interview, you should include some sample questions or indicate what the questions were about.
8.The procedure should be described more clearly and accurately.
9.It is not clear who did the categorisation of the contents, whether it was arrived at by inter-judge agreement, etc. The same applies to the coding of the contents.
10.In the results section, even if the research is qualitative, quantitative data can always be provided to clarify, for example, how many arguments were included in the different coding categories.
11.I think that with the results presented in the study it is very daring to discuss them, for me, it is a question of whether you believe it or not. I have no data to be able to judge whether or not the results obtained are in line with other research included in the paper.
1.The title does not reflect the true objective of the research, which is not the quality of nursing care, but rather the description of how certain nursing staff spent the COVID-19 pandemic. Therefore, I think they should change the title and adjust it to the real objective of the study.
2.Throughout the text, I believe that it confuses the obligatory professional care that must be dispensed by nurses with quality nursing care. This happens because they do not define what they mean by quality in nursing care (an aspect that does not appear in the introduction). From my point of view, when we analyse the quality of care, we must contrast it with other data taken previously or with previously established standards, as well as collecting data from the people on whom the nursing care and attention falls. To say that there has been quality in nursing care with the measures that have been collected and the subjects included in the research is, in my opinion, very adventurous and rather subjective.
3.I believe that the introduction does not go into the topics or variables that are then highlighted in the research in the results section. It should include the quality of nursing care, the results obtained in other research on fears of contagion, the stress caused by the COVID-19 situation, among others.
4.From my point of view, the objective should go at the end of the introduction or justification and not in the method.
5.The method section is very confusing. I consider that the information should be structured following the classic sections, participants, instruments and procedure.
6.When you talk about the participants, you should indicate the number of participants and whether or not they are representative of the sample, at least of the hospital. You should also clearly specify the sex of the participants although it may be obvious.
7.When talking about the instrument used, even if it is a semi-structured interview, you should include some sample questions or indicate what the questions were about.
8.The procedure should be described more clearly and accurately.
9.It is not clear who did the categorisation of the contents, whether it was arrived at by inter-judge agreement, etc. The same applies to the coding of the contents.
10.In the results section, even if the research is qualitative, quantitative data can always be provided to clarify, for example, how many arguments were included in the different coding categories.
11.I think that with the results presented in the study it is very daring to discuss them, for me, it is a question of whether you believe it or not. I have no data to be able to judge whether or not the results obtained are in line with other research included in the paper.
1.The title does not reflect the true objective of the research, which is not the quality of nursing care, but rather the description of how certain nursing staff spent the COVID-19 pandemic. Therefore, I think they should change the title and adjust it to the real objective of the study.
2.Throughout the text, I believe that it confuses the obligatory professional care that must be dispensed by nurses with quality nursing care. This happens because they do not define what they mean by quality in nursing care (an aspect that does not appear in the introduction). From my point of view, when we analyse the quality of care, we must contrast it with other data taken previously or with previously established standards, as well as collecting data from the people on whom the nursing care and attention falls. To say that there has been quality in nursing care with the measures that have been collected and the subjects included in the research is, in my opinion, very adventurous and rather subjective.
3.I believe that the introduction does not go into the topics or variables that are then highlighted in the research in the results section. It should include the quality of nursing care, the results obtained in other research on fears of contagion, the stress caused by the COVID-19 situation, among others.
4.From my point of view, the objective should go at the end of the introduction or justification and not in the method.
5.The method section is very confusing. I consider that the information should be structured following the classic sections, participants, instruments and procedure.
6.When you talk about the participants, you should indicate the number of participants and whether or not they are representative of the sample, at least of the hospital. You should also clearly specify the sex of the participants although it may be obvious.
7.When talking about the instrument used, even if it is a semi-structured interview, you should include some sample questions or indicate what the questions were about.
8.The procedure should be described more clearly and accurately.
9.It is not clear who did the categorisation of the contents, whether it was arrived at by inter-judge agreement, etc. The same applies to the coding of the contents.
10.In the results section, even if the research is qualitative, quantitative data can always be provided to clarify, for example, how many arguments were included in the different coding categories.
11.I think that with the results presented in the study it is very daring to discuss them, for me, it is a question of whether you believe it or not. I have no data to be able to judge whether or not the results obtained are in line with other research included in the paper.
1.The title does not reflect the true objective of the research, which is not the quality of nursing care, but rather the description of how certain nursing staff spent the COVID-19 pandemic. Therefore, I think they should change the title and adjust it to the real objective of the study.
2.Throughout the text, I believe that it confuses the obligatory professional care that must be dispensed by nurses with quality nursing care. This happens because they do not define what they mean by quality in nursing care (an aspect that does not appear in the introduction). From my point of view, when we analyse the quality of care, we must contrast it with other data taken previously or with previously established standards, as well as collecting data from the people on whom the nursing care and attention falls. To say that there has been quality in nursing care with the measures that have been collected and the subjects included in the research is, in my opinion, very adventurous and rather subjective.
3.I believe that the introduction does not go into the topics or variables that are then highlighted in the research in the results section. It should include the quality of nursing care, the results obtained in other research on fears of contagion, the stress caused by the COVID-19 situation, among others.
4.From my point of view, the objective should go at the end of the introduction or justification and not in the method.
5.The method section is very confusing. I consider that the information should be structured following the classic sections, participants, instruments and procedure.
6.When you talk about the participants, you should indicate the number of participants and whether or not they are representative of the sample, at least of the hospital. You should also clearly specify the sex of the participants although it may be obvious.
7.When talking about the instrument used, even if it is a semi-structured interview, you should include some sample questions or indicate what the questions were about.
8.The procedure should be described more clearly and accurately.
9.It is not clear who did the categorisation of the contents, whether it was arrived at by inter-judge agreement, etc. The same applies to the coding of the contents.
10.In the results section, even if the research is qualitative, quantitative data can always be provided to clarify, for example, how many arguments were included in the different coding categories.
11.I think that with the results presented in the study it is very daring to discuss them, for me, it is a question of whether you believe it or not. I have no data to be able to judge whether or not the results obtained are in line with other research included in the paper.
I consider that the article does not add much to the data and knowledge on the subject that has been studied in recent years. Furthermore, I believe that in order to be published, it needs to be restructured:
I consider that the article does not add much to the data and knowledge on the subject that has been studied in recent years. Furthermore, I believe that in order to be published, it needs to be restructured:
I consider that the article does not add much to the data and knowledge on the subject that has been studied in recent years. Furthermore, I believe that in order to be published, it needs to be restructured:
I consider that the article does not add much to the data and knowledge on the subject that has been studied in recent years. Furthermore, I believe that in order to be published, it needs to be restructured:
I consider that the article does not add much to the data and knowledge on the subject that has been studied in recent years. Furthermore, I believe that in order to be published, it needs to be restructured:
I consider that the article does not add much to the data and knowledge on the subject that has been studied in recent years. Furthermore, I believe that in order to be published, it needs to be restructured:
I consider that the article does not add much to the data and knowledge on the subject that has been studied in recent years. Furthermore, I believe that in order to be published, it needs to be restructured:
I consider that the article does not add much to the data and knowledge on the subject that has been studied in recent years. Furthermore, I believe that in order to be published, it needs to be restructured:
I consider that the article does not add much to the data and knowledge on the subject that has been studied in recent years. Furthermore, I believe that in order to be published, it needs to be restructured:
I consider that the article does not add much to the data and knowledge on the subject that has been studied in recent years. Furthermore, I believe that in order to be published, it needs to be restructured:
I consider that the article does not add much to the data and knowledge on the subject that has been studied in recent years. Furthermore, I believe that in order to be published, it needs to be restructured:
I consider that the article does not add much to the data and knowledge on the subject that has been studied in recent years. Furthermore, I believe that in order to be published, it needs to be restructured:
I consider that the article does not add much to the data and knowledge on the subject that has been studied in recent years. Furthermore, I believe that in order to be published, it needs to be restructured:
I consider that the article does not add much to the data and knowledge on the subject that has been studied in recent years. Furthermore, I believe that in order to be published, it needs to be restructured:
I consider that the article does not add much to the data and knowledge on the subject that has been studied in recent years. Furthermore, I believe that in order to be published, it needs to be restructured:
I consider that the article does not add much to the data and knowledge on the subject that has been studied in recent years. Furthermore, I believe that in order to be published, it needs to be restructured:
I consider that the article does not add much to the data and knowledge on the subject that has been studied in recent years. Furthermore, I believe that in order to be published, it needs to be restructured:
I consider that the article does not add much to the data and knowledge on the subject that has been studied in recent years. Furthermore, I believe that in order to be published, it needs to be restructured:
I consider that the article does not add much to the data and knowledge on the subject that has been studied in recent years. Furthermore, I believe that in order to be published, it needs to be restructured:
I consider that the article does not add much to the data and knowledge on the subject that has been studied in recent years. Furthermore, I believe that in order to be published, it needs to be restructured:
I consider that the article does not add much to the data and knowledge on the subject that has been studied in recent years. Furthermore, I believe that in order to be published, it needs to be restructured:
I consider that the article does not add much to the data and knowledge on the subject that has been studied in recent years. Furthermore, I believe that in order to be published, it needs to be restructured:
I consider that the article does not add much to the data and knowledge on the subject that has been studied in recent years. Furthermore, I believe that in order to be published, it needs to be restructured:
I consider that the article does not add much to the data and knowledge on the subject that has been studied in recent years. Furthermore, I believe that in order to be published, it needs to be restructured:
I consider that the article does not add much to the data and knowledge on the subject that has been studied in recent years. Furthermore, I believe that in order to be published, it needs to be restructured:
I consider that the article does not add much to the data and knowledge on the subject that has been studied in recent years. Furthermore, I believe that in order to be published, it needs to be restructured:
I consider that the article does not add much to the data and knowledge on the subject that has been studied in recent years. Furthermore, I believe that in order to be published, it needs to be restructured:
I consider that the article does not add much to the data and knowledge on the subject that has been studied in recent years. Furthermore, I believe that in order to be published, it needs to be restructured:

Round 2
Reviewer 3 Report
I believe that with the changes made to the article it could be published.
Cite
Stavropoulou, A.; Rovithis, M.; Sigala, E.; Moudatsou, M.; Fasoi, G.; Papageorgiou, D.; Koukouli, S. Exploring Nurses’ Working Experiences during the First Wave of COVID-19 Outbreak. Healthcare 2022, 10, 1406. https://doi.org/10.3390/healthcare10081406
Stavropoulou A, Rovithis M, Sigala E, Moudatsou M, Fasoi G, Papageorgiou D, Koukouli S. Exploring Nurses’ Working Experiences during the First Wave of COVID-19 Outbreak. Healthcare. 2022; 10(8):1406. https://doi.org/10.3390/healthcare10081406
Chicago/Turabian StyleStavropoulou, Areti, Michael Rovithis, Evangelia Sigala, Maria Moudatsou, Georgia Fasoi, Dimitris Papageorgiou, and Sofia Koukouli. 2022. "Exploring Nurses’ Working Experiences during the First Wave of COVID-19 Outbreak" Healthcare 10, no. 8: 1406. https://doi.org/10.3390/healthcare10081406
APA StyleStavropoulou, A., Rovithis, M., Sigala, E., Moudatsou, M., Fasoi, G., Papageorgiou, D., & Koukouli, S. (2022). Exploring Nurses’ Working Experiences during the First Wave of COVID-19 Outbreak. Healthcare, 10(8), 1406. https://doi.org/10.3390/healthcare10081406